# *as*-Indaceno[3,2,1,8,7,6-*ghijklm*]terrylene as a near-infrared absorbing C$_{70}$-fragment

Yuki Tanaka[1], Norihito Fukui [1✉] & Hiroshi Shinokubo [1✉]

Carbon and hydrogen are fundamental components of organic molecules and a fascinating plethora of functions can be generated using these two elements. Yet, realizing attractive electronic structures only by using carbon and hydrogen remains challenging. Herein, we report the synthesis and properties of the C$_{70}$ fragment *as*-indaceno[3,2,1,8,7,6-*ghijklm*]terrylene, which exhibits near-infrared (NIR) absorption (up to ca. 1300 nm), even though this molecule consists of only 34 carbon and 14 hydrogen atoms. A remarkably small highest occupied molecular orbital–lowest unoccupied molecular orbital (HOMO–LUMO) gap is confirmed by electrochemical measurement and theoretical calculations. Furthermore, *as*-indacenoterrylene is stable despite the absence of peripheral substituents, which contrasts with the cases of other NIR-absorbing hydrocarbons such as diradicaloids and antiaromatic molecules. The results of this study thus offer fundamental insights into the design of hydrocarbons with a small band gap.

[1] Department of Molecular and Macromolecular Chemistry, Graduate School of Engineering, Nagoya University, Nagoya 464-8603, Japan.
✉email: fukui@chembio.nagoya-u.ac.jp; hshino@chembio.nagoya-u.ac.jp

Bowl-shaped aromatic hydrocarbons have been actively explored in modern organic chemistry[1–8]. Owing to their structural nonplanarity, these molecules exhibit intriguing characteristics such as high solubility, structural flexibility[9–14], unique host–guest recognition[15–18] and high electronic conductivity[19,20]. Hence, endowing bowl-shaped aromatic hydrocarbons with unique electronic structures can be considered a promising approach to realize intriguing functions. Recently, Cao and co-workers[21] have reported indene-fused corannulene derivative **1**, which exhibits a narrow highest occupied molecular orbital–lowest unoccupied molecular orbital (HOMO–LUMO) gap due to its diradical nature (Fig. 1a). Such a nonplanar hydrocarbon with a small HOMO–LUMO gap is attractive with regard to potential applications including near-infrared (NIR)-absorbing dyes and ambipolar organic semiconductors[22–24].

During the past three decades, studies on bowl-shaped aromatic hydrocarbons have been related to $C_{60}$-fullerene fragments and their $\pi$-extended derivatives[25–30]. $C_{60}$ adopts a sphere-like structure with high symmetry ($I_h$), while $C_{70}$-fullerene shows lower symmetry ($D_{5h}$). Consequently, $C_{70}$ fragments are inherently diverse, albeit that the hitherto reported $C_{70}$ fragments remain limited[31–41]. Among these, especially dicyclopenta [4,3,2,1-*ghi*:4′,3′,2′,1′-*pqr*]perylene (**2**), synthesized by Wu and co-workers,[41] is worth mentioning, as **2** exhibits distinct antiaromaticity and a narrow HOMO–LUMO gap (Fig. 1b). Considering that antiaromatic compounds are relatively rare[42–45], this result highlights that the construction of $C_{70}$-fragments may offer a good opportunity to access unique bowl-shaped hydrocarbons with fascinating electronic structures.

Herein, we disclose the synthesis and properties of $C_{70}$ fragment *as*-indaceno[3,2,1,8,7,6-*ghijklm*]terrylene (**3**) (Fig. 1c). Surprisingly, **3** exhibits NIR absorption tailing to ca. 1300 nm although it contains only 34 carbons and 14 hydrogen atoms. *as*-Indacenoterrylene **3** is stable enough even without any peripheral substituents. This feature is contrasting with other NIR-absorbing hydrocarbons such as singlet diradicaloids and antiaromatic molecules.

## Results

### Synthesis of *as*-indacenoterrylene 3.

The synthesis of *as*-indacenoterrylene **3** is shown in Fig. 2a. The synthetic approach is based on the incorporation of $sp^3$-hybridized carbon atoms into the fused $\pi$-system to generate the curved structure. The bromine–lithium exchange reaction between 1-bromonaphthalene and butyllithium, followed by the addition of 1,4,5,8-tetrachloroanthraquinone (**4**), afforded the corresponding dinaphthylated diols. The crude mixture was methylated with iodomethane in the presence of sodium hydride, which provided a mixture of *cis*-adduct **5** and *trans*-adduct **6** in 22% and 64% yield (over two steps), respectively. These isomers exhibit different solubility in $CH_2Cl_2$ and were separated by filtration. The *cis*-isomer **5** engaged in a Pd-catalysed intramolecular C–H/C–Cl coupling[17] to provide methoxy-substituted *as*-indacenoterrylene **7** in 21% yield, wherein the two five-membered rings are located on the same side. Another isomer, in which the two five-membered rings are located on the opposite side, was not detected. The methoxy group in **7** was transformed to hydrogen using triethylsilane ($HSiEt_3$) and trifluoromethanesulfonic acid (TfOH)[46], which afforded **8** in 90% yield. The $^1H$ nuclear magnetic resonance (NMR) signal of the internal hydrogens was observed at 5.91 ppm (Supplementary Fig. 5), which is comparable to that of hydrofullerene $C_{60}H_2$ (5.93 ppm)[47]. Finally, abstraction of hydrogen from **8** using 1,4-dichloro-2,6-dicyano-*p*-benzoquinone (DDQ) furnished *as*-indacenoterrylene **3** in 73% yield. Although **3** exhibits very low solubility in most common organic solvents, it slightly dissolved in $CH_2Cl_2/CS_2$, *o*-dichlorobenzene and *N*-methylpyrrolidone. The attempted reductive aromatization of **7** with $SnCl_2$ and HCl resulted in decomposition without the formation of any isolable products. In addition, no fully fused product was detected in the reaction mixture when *trans*-adduct **6** was subjected to the intramolecular C–H/C–Cl coupling. This result could be due to the large structural distortion of the expected product having oppositely directed $sp^3$ carbons.

The structures of methoxy-substituted *as*-indacenoterrylene **7** and hydrogenated *as*-indacenoterrylene **8** were unequivocally determined by single-crystal X-ray diffraction analysis (Fig. 2b). Both **7** and **8** adopt a bowl-shaped structure with two $sp^3$-hybridized carbon atoms in the central six-membered ring. Notably, **7** and **8** can be regarded as internally functionalized bowl-shaped aromatics, which have been rarely explored[48–50]. The methoxy groups of **7** are aligned perpendicular to the $\pi$-surface, which shows a bowl-depth of 2.10 Å. Unfortunately, the crystal data of **8** are insufficient for a detailed structural analysis due to severe disorder.

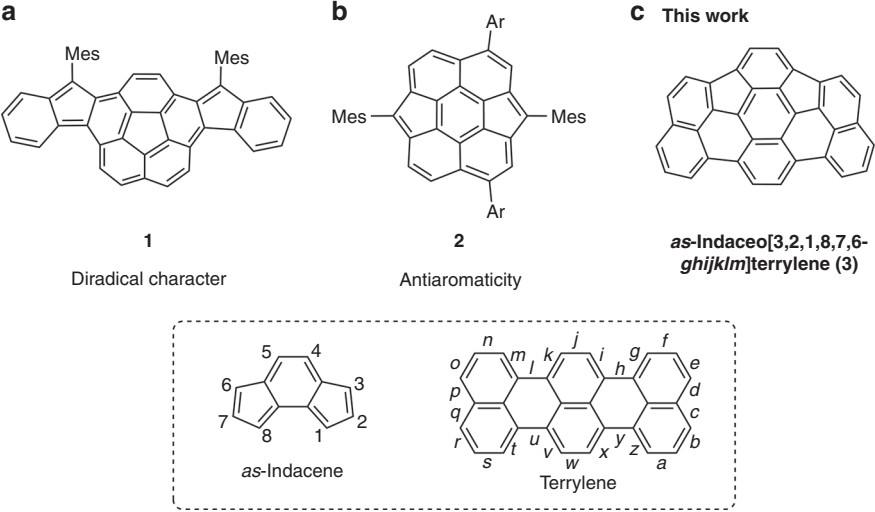

**Fig. 1 NIR-absorbing bowl-shaped (anti)aromatic hydrocarbons 1–3. a** Indene-fused corannulene derivative **1**. **b** Dicyclopenta[4,3,2,1-*ghi*:4′,3′,2′,1′-*pqr*] perylene (**2**). **c** *as*-Indaceno[3,2,1,8,7,6-*ghijklm*]terrylene (**3**).

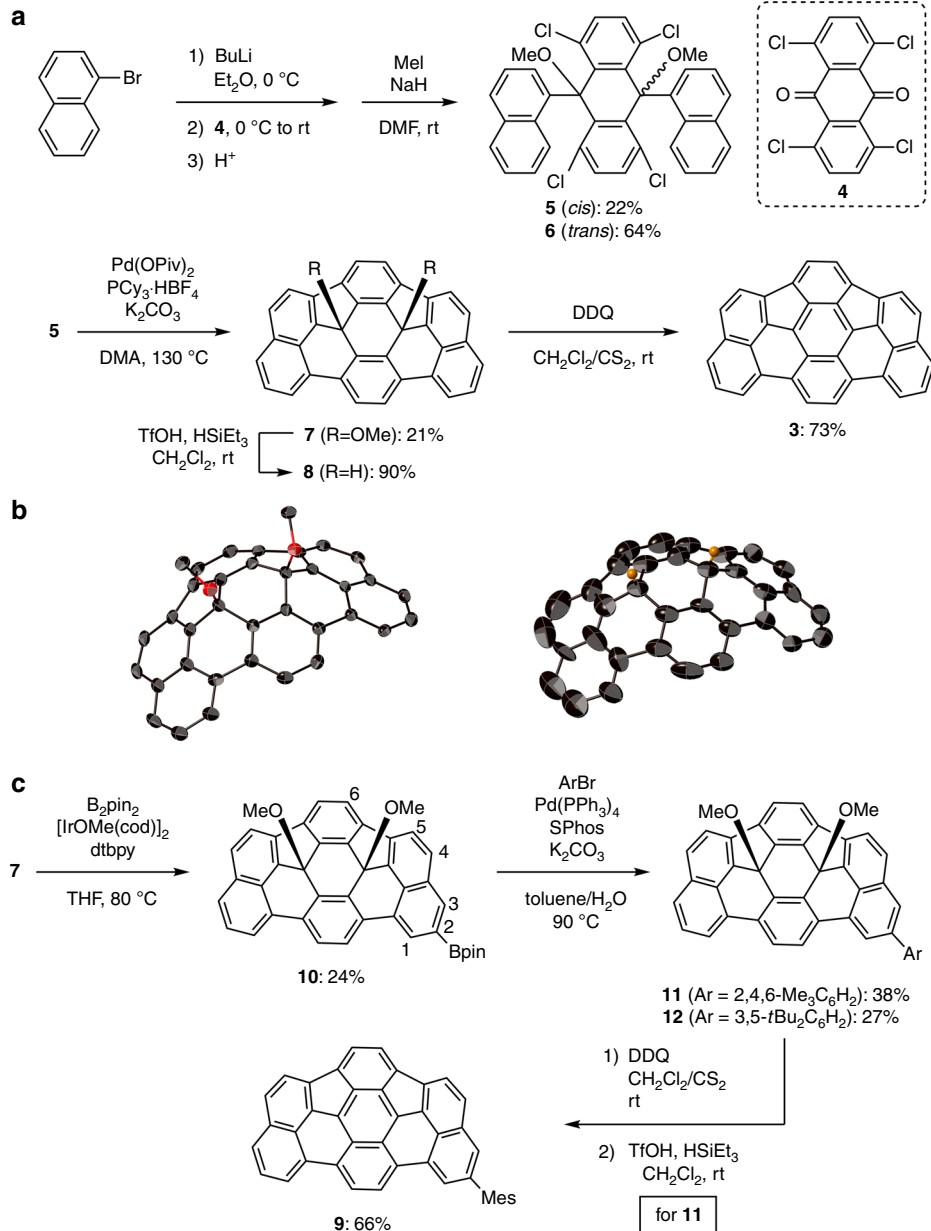

**Fig. 2 Syntheses and X-ray crystal structures. a** Synthesis of *as*-indacenoterrylene **3**. **b** X-ray crystal structures of internally methoxy-substituted *as*-indacenoterrylene **7** (left) and internally hydrogenated *as*-indacenoterrylene **8** (right). Thermal ellipsoids are drawn at 50% probability and all hydrogen atoms except for the internal hydrogen atoms of **8** are omitted for clarity. **c** Synthesis of 2-mesityl-*as*-indacenoterrylene **9**.

The $^1H$ NMR spectrum of *as*-indacenoterrylene **3** in CDCl$_3$/CS$_2$ exhibited seven signals in the range of 7.21–6.78 ppm, supporting a $C_s$-symmetric structure (Supplementary Fig. 7). These signals are upfield-shifted by ca. 1 ppm compared to those of other C$_{70}$ fragments, except for antiaromatic dicyclopentaperylene **2** (refs. [31–41]). The upfield-shifted $^1H$ NMR signals are characteristic for π-conjugated systems with narrow HOMO–LUMO gaps as theoretical studies suggest that, in such systems, the quantum-chemical perturbation of excited states to the ground state results in the increase of paratropic ring current[51,52]. The high-resolution atmospheric pressure chemical ionization time-of-flight (APCI-TOF) mass spectrum of **3** showed a parent ion peak at $m/z = 423.1184$ (calcd for C$_{34}$H$_{14}$, $m/z = 423.1168$ [M + H]$^+$), which is in accordance with the expected molecular composition.

As the low solubility of *as*-indacenoterrylene **3** hampered a detailed investigation of the physical properties, we synthesized 2-mesityl-*as*-indacenoterrylene **9** (Fig. 2c). The iridium-catalysed C–H borylation[53] of **7** with bis(pinacolato)diboron furnished 2-borylated **10**. A subsequent Suzuki–Miyaura cross-coupling reaction with 2-bromomesitylene and 1-bromo-3,5-di-*tert*-butyl-benzene afforded the arylated products **11** and **12**, respectively. The structure of **12** was determined by a single-crystal X-ray diffraction analysis (Supplementary Fig. 21). Mesityl-substituted derivative **11** was further transformed to 2-mesityl-*as*-indaceno-terrylene **9**, which exhibits high solubility in common organic solvents including CH$_2$Cl$_2$, toluene and tetrahydrofuran.

**Bowl-to-bowl inversion.** Variable-temperature $^1H$ NMR spectra of **9** were recorded in CDCl$_3$ (Supplementary Fig. 22). At 0 °C, the signals that arise from the *ortho*-methyl groups of the mesityl substituent were observed as a pair of singlets. At 35 °C, these singlets coalesced to a single singlet at 2.08 ppm. A plot of the

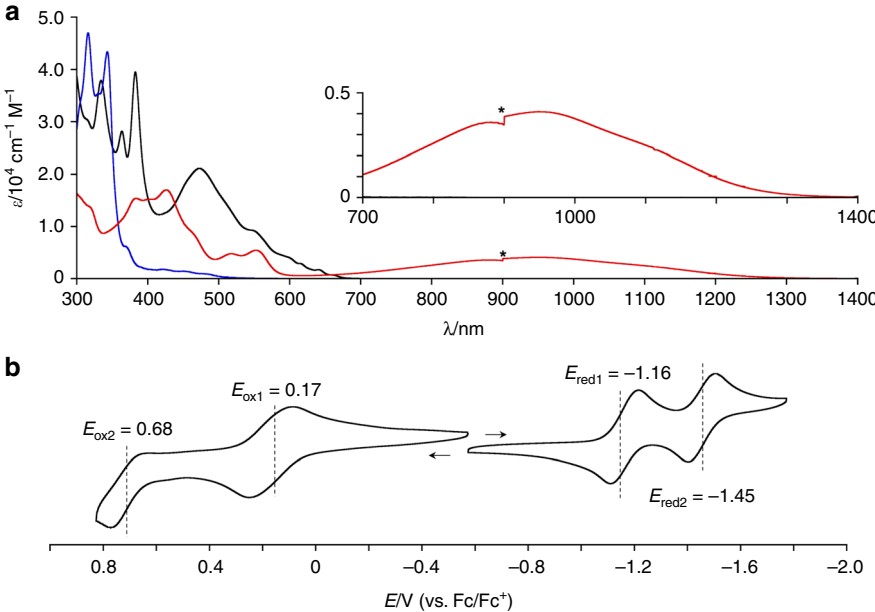

**Fig. 3 Narrow HOMO–LUMO gap of *as*-indacenoterrylenes. a** UV/vis/NIR absorption spectra of **3** (red), **7** (blue), and fullerene $C_{70}$ (black) in *o*-dichlorobenzene ($\lambda$: wavelength; $\varepsilon$: extinction coefficient). Noise was generated at 900 nm by switching detectors. **b** Cyclic voltammogram of 2-mesityl-*as*-indacenoterrylene **9** (solvent: $CH_2Cl_2$; supporting electrolyte: 0.1 M [Bu₄N][PF₆]; reference electrode: Ag/AgNO₃). The second oxidation potential was determined by differential pulse voltammetry.

experimental $\ln(k/T)$ values versus $1/T$ was fitted by the Eyring equation[54], affording the activation enthalpy $\Delta H^{\ddagger}$ and entropy $\Delta S^{\ddagger}$ of 16.9 kcal mol⁻¹ and 4.08 cal K⁻¹ mol⁻¹, respectively (Supplementary Fig. 23). Density functional theory (DFT) calculations at the CAM-B3LYP/6–311+G(d,p)//B3LYP/6–31G(d) level of theory suggest that the bowl-to-bowl inversion passes through a planar transition state (TS) with the activation barrier of 15.1 kcal mol⁻¹ (Supplementary Fig. 24). This value is smaller than the calculated rotation barrier of the peripheral mesityl group (26.1 kcal mol⁻¹) (Supplementary Fig. 25). Hence, the observed temperature dependence of NMR spectra can be interpreted in terms of an inversion of the bowl-shaped structure.

**Electronic structures.** The ultraviolet (UV)/vis/NIR absorption spectra of **3**, **7** and fullerene $C_{70}$ are shown in Fig. 3a. Methoxy-substituted **7** exhibits weak absorption bands in the range of 400–520 nm, and the presence of vibrational bands implies structural rigidity. The absorption of **7** is blueshifted compared to that of fullerene $C_{70}$ due to the disrupted conjugation on account of the central $sp^3$-hybridized carbon atoms. In sharp contrast, *as*-indacenoterrylene **3** displays a significantly redshifted absorption with a peak top at 952 nm and a tail to ca. 1300 nm. The optical HOMO–LUMO gap is 1.30 eV. The extinction coefficient of the broad absorption band ($4.1 \times 10^3$ cm⁻¹ M⁻¹) is considerable in comparison to those of the forbidden absorption of indene-fused corannulene derivative **1** (ref. [21]) and dicyclopentaperylene **2** (ref. [41]). As the forbidden $S_0$–$S_1$ transition is a diagnostic feature of antiaromatic molecules[42–45], the allowed nature of the low-energy absorption of **3** cannot be explained in terms of a potential contribution of antiaromaticity. Interestingly, the absorption tail of **3** is even comparable to that of quarteranthene[55], which is a zigzag-edge nanocarbon that consists of 56 carbon atoms. The NIR absorption band of **3** obeys the Lambert–Beer's law and was matched with that of mesityl-substituted derivative **9** (Supplementary Fig. 26). These results clearly suggest a negligible effect of aggregation on the NIR absorption.

The electrochemical properties of 2-mesityl-*as*-indacenoterrylene **9** were studied by cyclic voltammetry (Fig. 3b), using the ferrocene/ferrocenium couple as an external reference, which revealed one reversible oxidation wave at 0.17 V as well as two reversible reduction waves at –1.16 and –1.45 V. The electrochemical HOMO–LUMO gap ($\Delta E = 1.33$ eV) is in good agreement with the optical HOMO–LUMO gap. Indene-fused corannulene derivative **1** exhibited redox potentials $E_{ox1}$ and $E_{red1}$ at 0.33 and –0.81 V, respectively[21]. Dicyclopentaperylene **2** displayed redox potentials $E_{ox1}$ and $E_{red1}$ at 0.33 and –0.96 V, respectively[41]. The $\Delta E$ value of **9** is slightly wider than that of **1** and comparable to that of **2**. The redox potentials of **9** are by ca. 0.2–0.3 V positively shifted relative to those of **1** and **2**, which is indicative of a higher electron-donating ability.

To evaluate the electronic structure of **3**, DFT calculations were conducted at the CAM-B3LYP/6–311+G(d,p) level of theory (Supplementary Figs. 27 and 29). Time-dependent (TD)-DFT calculations predicted the presence of the HOMO–LUMO transition at 820 nm with an oscillator strength of 0.1226, which is in good agreement with the experimental absorption spectrum (Supplementary Fig. 28, Supplementary Table 1). The calculated HOMOs and LUMOs of **3** and its fragments, *as*-indacenopyrene and terrylene, are shown in Fig. 4a and Supplementary Fig. 31. The HOMO of **3** is delocalized along the longitudinal axis, whose distribution and energy level are almost identical to those of the HOMO of terrylene rather than those of HOMO of *as*-indacenopyrene. In contrast, the LUMO of **3** is mainly located at the centre of the molecule, which resembles the LUMO of *as*-indacenopyrene and energetically much lower than the LUMO of terrylene by 0.7 eV. The high electron-deficiency of *as*-indaceno-pyrene can be attributed to the incorporated non-alternant hydrocarbon segments[56].

Nucleus-independent chemical shift (NICS)[57–59] calculations and anisotropy of the induced current density (ACID)[60] calculations were conducted for *as*-indacenoterrylene **3** at the B3LYP/6–31G(d) level (Fig. 4c, d). The NICS(0)$_{iso}$ values, except for that at the five-membered ring, fall in the range –3.9 to 6.4 ppm. An essentially similar trend was also observed for the NICS(1)$_{zz}$ values (Fig. 4c, Supplementary Table 2)[59]. The ACID plot indicates a negligible contribution of the macrocyclic ring current.

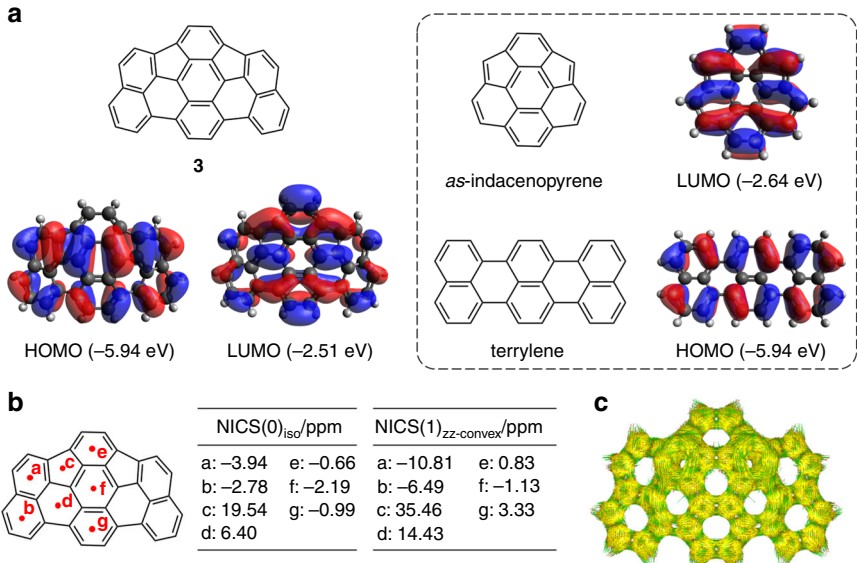

**Fig. 4 Computational results for the electronic structure of *as*-indacenoterrylene 3. a** HOMO and LUMO of **3**. Inset: HOMO of terrylene and LUMO of *as*-indacenopyrene. **b** NICS values of **3**. **c** ACID plot of **3**.

Besides, the ACID calculations predict counter-clockwise ring current at the five-membered rings, which accord with the positively large NICS values. Although the exact reason remains unclear, this observation can be accounted for by the local antiaromaticity at the five-membered rings, which could be due to the cyclopentadienyl cation-like electronic contribution associated with the electron-deficient character of *as*-indacenopyrene segment. These results suggest that the overall electronic structure of **3** cannot be rationalized satisfactorily in term of antiaromaticity.

Calculations of **3** at the CASSCF(2,2)/6–31G level of theory furnished a diradical index ($y = 0.05$) that is comparable to that of bisanthene ($y = 0.07$) and much smaller than that of teranthene ($y = 0.54$)[55]. The increase of the calculation level (CASSCF(4,6)/6–31G(d)) again provided a relatively small diradical index of $y = 0.07$. In addition, the closed-shell (CS) singlet, open-shell (OS) singlet and triplet states of **3** were optimized at the (U)CAM-B3LYP/6–311+G(d,p) level (Supplementary Tables 3–5). The OS singlet state is structurally almost identical with the CS singlet state, being slightly more stable than the CS singlet state by 0.9 kcal mol$^{-1}$ with the small diradical index of $y = 0.03$. The triplet state is energetically unfavourable with the energy gap $\Delta E_{ST}$ of 6.7 kcal mol$^{-1}$. These results suggest the non-dominant contribution of an OS diradical state, which will be due to a small energetical gain given by the formation of one Clar's sextet in its resonance structure (Supplementary Fig. 36)[61]. This assumption has been supported by the experimental fact that heating an *o*-dichlorobenzene solution of **3** at 100 °C for 12 h in air resulted in no distinct spectral change of the absorption, indicating that **3** is remarkably stable despite the absence of peripheral substituents (Supplementary Fig. 35).

The above-mentioned studies indicate that the electronic structure of *as*-indacenoterrylene **3** cannot be satisfactorily explained in terms of its antiaromaticity and diradical character. One plausible description could be the coexistence of *as*-indacenopyrene- and terrylene units, which are moderately merged together without diminishing the individual electronic character. TD-DFT calculations predict that, while the $S_0$–$S_1$ transition of terrylene is allowed (oscillator strength = 0.8363) with a large gap of 2.52 eV, that of *as*-indacenopyrene is forbidden (oscillator strength = 0.0118) with a small gap of

0.97 eV (Supplementary Figs. 33 and 34). The oscillator strength and energy gap of **3** (0.1226 and 1.51 eV) lie between those of terrylene and *as*-indacenopyrene. Importantly, the transition dipoles of **3**, *as*-indacenopyrene and terrylene are directed in the same molecular axis, which could be the origin of the moderate mixing of the transitions. Because of the coexistence of electron-accepting and electron-donating segments[62], the $S_0$–$S_1$ transition of **3** could include some contribution of the charge-transfer (CT) character. Indeed, the electrostatic potential maps suggest that the $S_0$–$S_1$ transition associates with the change of orbital- and charge-distributions (Supplementary Figs. 30 and 32). The ill-defined absorption feature of **3** and **9** with the absence of vibrational bands in the NIR region is reminiscent of CT character.

## Discussion

We have synthesized the $C_{70}$ fragment *as*-indaceno[3,2,1,8,7,6-*ghijklm*]terrylene **3**. The synthetic concept is based on the generation of curvature by incorporation of $sp^3$-hybridized carbon atoms in the fused π-system. The structure was assigned based on NMR spectroscopy and mass spectrometry. The physical properties were probed by UV/vis/NIR absorption spectroscopy, cyclic voltammetry, temperature-dependent NMR analysis, and theoretical calculations. Even though **3** contains only 34 carbon and 14 hydrogen atoms, it absorbs in the NIR region up to ca. 1300 nm. The absorption tail of **3** is comparable to organic diradicaloid **1** and antiaromatic hydrocarbon **2**. However, *as*-indacenoterrylene **3** exhibits outstanding stability despite the absence of peripheral substituents, which stands in sharp contrast to the behaviour of other NIR-absorbing hydrocarbons such as diradicaloids and antiaromatic molecules. The current study thus offers effective design guidelines for aromatic hydrocarbons with narrow HOMO–LUMO gaps.

## Methods

**Materials and characterization.** $^1$H NMR (500 MHz) and $^{13}$C NMR (126 MHz) spectra were recorded on a Bruker AVANCE III HD spectrometer. Chemical shifts were reported as the delta scale in ppm relative to CDCl$_3$ ($\delta = 7.26$) for $^1$H NMR and CDCl$_3$ ($\delta = 77.16$ ppm) for $^{13}$C NMR. The $^1$H and $^{13}$C NMR spectra of structurally novel chemical compounds are shown in Supplementary Figs. 1–17. Mass spectra were recorded using a Bruker microTOF by APCI methods. UV/vis/NIR absorption spectra were recorded on a Shimadzu UV-2550 or JASCO V670 spectrometer. X-ray data were obtained using a Rigaku CCD diffractometer

(Saturn 724 with MicroMax-007) with Varimax Mo optics using graphite mono-chromated Mo-Kα radiation (λ = 0.71075 Å). The crystal structures of **5**, **7**, **8** and **12** are shown in Supplementary Figs. 18–21. Redox potentials were measured by the cyclic voltammetry method on an ALS electrochemical analyzer model 612C. Unless otherwise noted, materials obtained from commercial suppliers were without further purification.

**Synthesis**. Selected procedures are shown below. Other procedures as well as compound data are described in the Supplementary Methods.

*Dimethoxy-substituted dihydro-as-indaceno[3,2,1,8,7,6-ghijklm]terrylene* (**7**). A Schlenk tube containing **5** (100 mg, 0.154 mmol), Pd(OPiv)$_2$ (36.8 mg, 0.119 mmol), PCy$_3$•HBF$_4$ (87.6 mg, 0.238 mmol) and K$_2$CO$_3$ (220 mg, 1.59 mmol) was evacuated and then refilled with N$_2$. To the tube, dry and degassed *N,N*-dimethylacetamide (DMA) (6.0 ml) was added. The mixture was stirred at 130 °C for 2 h and then poured into CH$_2$Cl$_2$. The mixture was passed through Celite. After removing the solvent, the purification by silica-gel column chromatography (CH$_2$Cl$_2$ as eluent) afforded **7** in 21% yield (16.5 mg, 34.0 μmol) as a yellow solid.

*Dihydro-as-indaceno[3,2,1,8,7,6-ghijklm]terrylene* (**8**). A 50 mL two-necked round-bottom flask containing **7** (33.4 mg, 68.9 mmol) was evacuated and then refilled with N$_2$. To the tube, CH$_2$Cl$_2$ (40 mL, dried and degassed) and triethylsilane (414 mg, 3.56 mmol) were added. After stirring, trifluoromethanesulfonic acid (51.6 mg, 0.344 mmol) was added. The mixture was stirred at room temperature for 1.5 h. The reaction was quenched with NEt$_3$ (25.3 mg, 0.250 mmol). The mixture was passed through short silica-gel (CH$_2$Cl$_2$ as an eluent). The solvent was removed by rotary evaporator. Washing the residue with MeOH and hexane afforded **8** in 90% yield (26.2 mg, 61.7 mmol) as a yellow solid.

*as-Indaceno[3,2,1,8,7,6-ghijklm]terrylene* (**3**). To a 50 ml round-bottom flask, **8** (11.7 mg, 27.6 μmol), CH$_2$Cl$_2$ (15 mL) and CS$_2$ (15 mL) were added. After stirring, a solution of DDQ in CH$_2$Cl$_2$ (6.21 mM, 8.30 mL) was added. The mixture was stirred at room temperature for 30 min. Saturated NaHCO$_3$ aq. was added to the mixture. The reaction mixture was extracted with CH$_2$Cl$_2$/CS$_2$ (1/1). The combined organic layers were dried over Na$_2$SO$_4$ and passed through short silica-gel (CH$_2$Cl$_2$ as an eluent). After removing the solvent, the purification by washing with hexane, MeOH and CHCl$_3$ afforded **3** in 73% yield (8.50 mg, 20.1 μmol) as a dark red solid.

**Theoretical calculations**. All calculations were carried out using the *Gaussian 09* program[63]. Initial geometry of **3** was obtained from the X-ray structure of **7**. The gas phase S$_0$- and S$_1$-geometries of **3** were optimized with the DFT method at the CAM-B3LYP/6–311+G(d,p) level[64]. Molecular orbitals were calculated at the CAM-B3LYP/6–311+G(d,p) level. The oscillator strengths of **3** were computed by the TD-DFT formalism at the CAM-B3LYP/6–311+G(d,p) level. The NICS and ACID calculations were conducted at the B3LYP/6–31G(d) level.

For the calculations of the possible diradical contribution, the geometries of **3** with the CS singlet, OS singlet and triplet states were optimized at the (U)CAM-B3LYP/6–311+G(d,p) level. The stability analysis was conducted to check the stability of the OS singlet solution. The diradical index $y$ was obtained on the basis of the natural orbital occupation number (NOON) calculation. We also conducted CASSCF calculations at the CS optimized structure to determine $y$ at the CASSCF(2,2)/6–31G and CASSCF(4,6)/6–31G(d) levels.

The TS geometry and intrinsic reaction coordinate were calculated at the B3LYP/6–31G(d) level by using the GRRM17 program[65]. The energy and freq calculations of the TS and ground state geometries were conducted at the CAM-B3LYP/6–311+G(d,p) level.

## Data availability

Crystallographic data (CIF files) for **5**, **7**, **8** and **12** have been deposited with the Cambridge Crystallographic Data Centre as supplementary publications. CCDC 2004717 (**5**), CCDC 2004720 (**7**), CCDC 2004719 (**8**) and CCDC 2004718 (**12**) contain the supplementary crystallographic data. These data can be obtained free of charge from the Cambridge Crystallographic Data Centre via [www.ccdc.cam.ac.uk/data_request/cif]. All other data supporting the findings of this study are available within the article and its Supplementary Information.

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

## Acknowledgements
This work was supported by JSPS KAKENHI grants JP26102003, JP15K21721, JP17H01190 and JP20K15257.

## Author contributions
H.S. and N.F. designed and conducted the project and prepared the manuscript. Y.T. carried out the synthesis and characterization.

## Competing interests
The authors declare no competing interests.
