## [Peer Review File · Nature Communications]

REVIEWER COMMENTS

Reviewer #1 (Remarks to the Author):

This is a very nice report on a new bowl-shaped C70 fragment. The authors developed efficient method to synthesize a large size (C34) as-indacenoterrylene, which displays unusual NIR absorption up to 1300 nm. It is a surprise to me to see that a medium-sized hydrocarbon can exhibit such long-wavelength absorption. The small energy gap is ascribed to intramolecular charge transfer according to TD DFT calculations, rather than the diradical character or anti-aromaticity. The chemistry likely can be extended to other low bandgap PAHs. I would like to strongly recommend acceptance of this work in Nature Communications. The paper can be further improved by considering the followings:

- (1) The bowl inversion energy barrier was experimentally estimated by VT NMR by using the line-shape analysis. It is advised to give the detailed analysis including the thermodynamic parameters (ΔH and ΔS) in the Supplementary Materials.
- (2) In the abstract, the authors claimed "Nevertheless, as-indacenoterrylene is surprisingly stable despite the absence of peripheral substituents, ...". Maybe "surprisingly" can be deleted as this is still an aromatic system, and I do not expect poor stability.
- (3) Figure 2, the last step for the synthesis of compound 9, shouldn't the hydrogenation take place first and then oxidative dehydrogenation second? Did you try to perform reduction of the methoxy-precursors by using such as SnCl₂ (with acid) directly?
- (4) Chemical oxidation/reduction followed by UV-vis-NIR and NMR could be interesting considering of the amphoteric properties of 9. But this can be done later given the current COVID-19 lockdown situation. It is NOT necessary to include in the current report.

Reviewer #2 (Remarks to the Author):

This is an interesting manuscript which describes a family of molecular nanographenes derived from as-indaceno[3,2,1,8,7,6-ghijklm]terrylene. Although the final structure 3 has not been confirmed by X-ray diffraction experiments, including the case of more soluble compound 9, these molecules have undoubtedly been characterized by NMR and mass spectrometry. Interestingly, unsubstituted derivative 3 shows a low absorption band in the NIR region attributed to an intramolecular charge transfer between terrylene fragment HOMO and as-indacenopyrene LUMO, that are mixed together in the carbon framework of the molecule. This novelty contrasts with previous examples where NIR absorption in PAHs was attributed to antiaromatic or diradical examples. The synthetic approach to induce curvature in nanographenes and the unprecedented NIR absorption of this type of systems are reasons enough to suggest the acceptance of this manuscript in Nature Communications after addressing the following concerns:

1. The synthesis of compound 3 was restricted by the poor stereoselectivity of the first one-pot step that only yields a 22% of the cis compound 5 used to continue the synthetic procedure. What happened with trans diastereomer 6 (64%) in the C-H/C-Cl intramolecular coupling? This information would be of interest for the readership.
2. Authors ensure that the upfield-shifted ¹H NMR signals of compound 3 compared with other C70 fragments is characteristic of π -conjugated systems with narrow HOMO-LUMO gap. It is no clear, however, the relationship between the NMR shift and the HOMO-LUMO gap. Please, clarify.
3. Authors described the bowl to bowl inversion of molecule 9 at 35 °C. In the ¹H NMR of the mesityl derivative 9, the coalescence of the methyl groups can be observed at this temperature. Based on the theoretical calculations, this is evidence to assume the bowl to bowl inversion. Could the coalescence of this signal be due to the rotation barrier of the mesityl group, as suggest the coalescence of the o-methyl groups in a unique signal?

4. The biradical structure for compound 3 would provide a structure with four pi-sextets compared with the neutral compound having three pi-sextets. According to Clar's rule, the diradical would be a plausible system (see figure). However, the ¹H NMR spectra of 3 does not show any significant broadening of the signals. Nevertheless, a further comment could be given in the main manuscript.

5. The experimental findings match pretty well with the theoretical calculations. In particular, the electrostatic potential shows the charge distribution along the molecule. Furthermore, in the ACID study the paratropic current of five member rings is also observed. However, a comment on this fact should be added to the discussion.

6. In addition to the isotropic NICS(0), calculation of NICS(1)zz (out-of-plane tensor component computed 1 Å above), since typically performs better than isotropic NICS(0). See: H. Fallah-Bagher-Shaidaei, C. S. Wannere, C. Corminboeuf, R. Puchta and P. v. R. Schleyer, *Org. Lett.*, 2006, 8, 863).

Other observations:

- In figure 2 parts (a, b, c) are not clearly separated.

- Some NMR spectra in the supporting information are not intense enough to determine the purity of the compounds. Despite the spectra have been amplified, but only the aromatic region.

- In the first introducing paragraph some references are provided but some recent ones, as claimed, should be added. See, for instance: *J. Am. Chem. Soc.* 2018, 140, 17188-17196; *Angew. Chem. Int. Ed.* 2017, 56, 9003-9007; for a recent review: *Commun Chem* 2019, 2, 58). The same for supramolecular interactions, see, for instance: *Nat. Commun.* 2015, 6:8215; *Angew. Chem. Int. Ed.* 2014, 53, 2170-2175.

Reviewer #3 (Remarks to the Author):

Review of NCOMMS-20-15443-T

The submitted work by Takana, Fukui and Shinokubo presents the synthesis and characterization of as-indacenoterrylene. This molecular system is stable and exhibits a low HOMO-LUMO gap due to intramolecular charge transfer (CT). The work provides novel avenues for the design of stable hydrocarbons with appealing optoelectronic properties.

In general, the manuscript is well written, clear and properly organized. In my opinion, this is a nice piece of work that deserves the attention of the Nature Communications' readers. Despite that, several issues need addressed before I can recommend it for publication.

Concretely, I do have two main concerns:

1. The authors claim that the near-infrared absorption band corresponds to an intramolecular charge transfer (CT) excitation, but in my opinion the nature of this transition needs to be further justified. Looking at the main orbital contribution (HOMO-to-LUMO in Table S1) and the large delocalization of the orbitals (Figures 4 and S27), it does not seem to me that there is a strong CT character. Moreover, electrostatic potential map (Figure 4) does not provide an unambiguous proof of the nature of the transition. This is a central element of the study and one of the main conclusions, thus, additional evidence for the (intra) CT character should be provided. For instance, the solvent polarity dependence of the absorption band might be an indication of CT. In

parallel, the authors should present computational evidence of the CT nature. Simple comparison between Mulliken, NPA and/or other schemes should confirm different ground and excited state charge localization. Representation of natural transition orbitals, or more sophisticated analysis based on attachment/detachment densities [J. Am. Chem. Soc. 131 (2009) 3913-3922], the transition density matrix [J. Chem. Theory Comput. 8 (2012) 2777-2789] or the hole/electron overlap [J. Chem. Phys. 128 (2008) 044118] and/or others could be also employed.

2. On the other hand, the authors suggest that, despite the small HOMO-LUMO gap, as-indacenoterrylene holds no diradical character. This conclusion is based (in part) on CASSCF(2,2)/6-31G calculations. This level of computation is rather low, especially considering the size of the molecule, which allows the use of higher order approximations. Is it possible to increase the active space of the calculation and the basis set at least to include polarization functions (6-31G*)?

Moreover, as far as I understand, these calculations have been performed on a closed-shell (CS) optimized geometry, as obtained with restricted Kohn-Sham (RKS) calculations, which sometimes might impose a CS solution (even at CASSCF level). Hence, in order to solidify the paper conclusions, I would suggest to try to optimize the open-shell (OS) solution and the triplet state. What is the OS/CS gap? And the relative energy to the lowest triplet? What is the diradical index (γ) at the OS geometry (if found)?

Stability analysis of the CS solution with respect to the OS singlet (e.g., RKS vs. UKS), might also help.

Minor issues:

Page 9. Remove DFT from the sentence: "DFT calculations of 3 at the CASSCF(2,2)/6-31G level".

Page 10. Figure 4. The authors show the similarities between the HOMO of 3 and terrylene, and the match between the LUMO of 3 and as-indacenopyrene. This is actually a very nice and simple comparison. But, I'm not sure it can be associated to a CT transition. What about the LUMO of terrylene? Does it resemble the LUMO of 3 as well? And the HOMO of as-indacenopyrene?

Response to Reviewers

Reviewer #1

This is a very nice report on a new bowl-shaped C₇₀ fragment. The authors developed efficient method to synthesize a large size (C₃₄) as-indacenoterrylene, which displays unusual NIR absorption up to 1300 nm. It is a surprise to me to see that a medium-sized hydrocarbon can exhibit such long-wavelength absorption. The small energy gap is ascribed to intramolecular charge transfer according to TD DFT calculations, rather than the diradical character or anti-aromaticity. The chemistry likely can be extended to other low bandgap PAHs. I would like to strongly recommend acceptance of this work in Nature Communications.

Response: We express our sincere thanks to this reviewer for his/her extensive reading and we are so delighted to see these supportive comments.

(1) The bowl inversion energy barrier was experimentally estimated by VT NMR by using the line-shape analysis. It is advised to give the detailed analysis including the thermodynamic parameters (ΔH^\ddagger and ΔS^\ddagger) in the Supplementary Materials.

Response: We thank this reviewer for his/her fruitful suggestion. We conducted the line-shape analysis, which provided the the activation enthalpy ΔH^\ddagger (16.9 kcal mol⁻¹) and entropy ΔS^\ddagger (4.08 cal K⁻¹ mol⁻¹). These results have been included in the main text (page 7, line 27) and SI (Figure S23) as follows.

A plot of the experimental $\ln(k/T)$ values versus $1/T$ was fitted by the Eyring equation⁵⁵, affording the activation enthalpy ΔH^\ddagger and entropy ΔS^\ddagger of 16.9 kcal mol⁻¹ and 4.08 cal K⁻¹ mol⁻¹, respectively (Figure S23).

Figure S23. Eyring plot for determination of the bowl-to-bowl inversion barrier.

(2) In the abstract, the authors claimed “Nevertheless, as-indacenoterrylene is surprisingly stable despite the absence of peripheral substituents, ...”. Maybe “surprisingly” can be deleted as this is still an aromatic system, and I do not expect poor stability.

Response: We appreciate the comment from this reviewer. The term "surprisingly" was removed from this sentence. In addition, the term "Nevertheless" was replaced with "Furthermore".

(3) Figure 2, the last step for the synthesis of compound **9**, shouldn't the hydrogenation take place first and then oxidative dehydrogenation second? Did you try to perform reduction of the methoxy-precursors by using such as SnCl₂ (with acid) directly?

Response: We thank this reviewer for his/her constructive suggestion. We conducted the reductive aromatization of methoxy precursor **7** with SnCl₂ and HCl. However, the reaction resulted in decomposition without the formation of any isolable products. This description has been added to the main text (page 4, line 20) as follows.

The attempted reductive aromatization of **7** with SnCl₂ and HCl resulted in decomposition without the formation of any isolable products.

(4) Chemical oxidation/reduction followed by UV-vis-NIR and NMR could be interesting considering of the amphoteric properties of **9**. But this can be done later given the current COVID-19 lockdown situation. It is NOT necessary to include in the current report.

Response: We thank this reviewer for his/her useful suggestion. We totally agree with this opinion, being aware of the importance of the redox chemistry of bowl-shaped hydrocarbons. We will explore this topic in due course as suggested by this reviewer. We would like to report the results in future.

Reviewer #2

This is an interesting manuscript which describes a family of molecular nanographenes derived from as-indaceno[3,2,1,8,7,6-ghijklm]terrylene. Although the final structure **3** has not been confirmed by X-ray diffraction experiments, including the case of more soluble compound **9**, these molecules have undoubtedly been characterized by NMR and mass spectrometry. Interestingly, unsubstituted derivative **3** shows a low absorption band in the NIR region attributed to an intramolecular charge transfer between terrylene fragment HOMO and as-indacenopyrene LUMO, that are mixed together in the carbon framework of

the molecule. This novelty contrasts with previous examples where NIR absorption in PAHs was attributed to antiaromatic or diradical examples. The synthetic approach to induce curvature in nanographenes and the unprecedented NIR absorption of this type of systems are reasons enough to suggest the acceptance of this manuscript in Nature Communications after addressing the following concerns:

Response: We express our sincere thanks to this reviewer for his/her extensive reading and we are so delighted to see these supportive comments.

1. The synthesis of compound **3** was restricted by the poor stereoselectivity of the first one-pot step that only yields a 22% of the cis compound **5** used to continue the synthetic procedure. What happened with trans diastereomer **6** (64%) in the C–H/C–Cl intramolecular coupling? This information would be of interest for the readership.

Response: We appreciate the important comment from this reviewer. *trans*-Adduct **6** was subjected to the intramolecular C–H/C–Cl coupling. However, we did not find a clue about the formation of a fully fused product, which will be due to the large structural distortion of the expected product having oppositely directed sp³ carbons. This description has been added to the main text (page 4, line 22) as follows.

In addition, no fully fused product was detected in the reaction mixture when *trans*-adduct **6** was subjected to the intramolecular C–H/C–Cl coupling. This result could be due to the large structural distortion of the expected product having oppositely directed sp³ carbons.

2. Authors ensure that the upfield-shifted ¹H NMR signals of compound **3** compared with other C₇₀ fragments is characteristic of π-conjugated systems with narrow HOMO–LUMO gap. It is no clear, however, the relationship between the NMR shift and the HOMO–LUMO gap. Please, clarify.

Response: We apologize for the lack of sufficient explanation. Theoretical studies (ref. 52 and 53) suggest that the narrow HOMO–LUMO gap results in the increase of paratropic ring current, which can be attributed to the quantum-chemical perturbation of excited states to the ground state. The related sentence in the main text has been modified (page 7, line 7) as follows.

The upfield-shifted ¹H NMR signals are characteristic for π-conjugated systems with narrow HOMO–LUMO gaps as theoretical studies suggest that, in such systems, the

quantum-chemical perturbation of excited states to the ground state results in the increase of paratropic ring current^{52,53}.

3. Authors described the bowl to bowl inversion of molecule **9** at 35 °C. In the ¹H NMR of the mesityl derivative **9**, the coalescence of the methyl groups can be observed at this temperature. Based on the theoretical calculations, this is evidence to assume the bowl to bowl inversion. Could the coalescence of this signal be due to the rotation barrier of the mesityl group, as suggest the coalescence of the *o*-methyl groups in a unique signal?

Response: The activation barrier of the rotation of the mesityl group was calculated, which gave a large activation barrier of 26.1 kcal mol⁻¹. Consequently, the observed temperature dependence of NMR spectra can be attributed to the bowl-to-bowl inversion. This explanation has been added to the main text (page 8, line 4) as follows.

Density functional theory (DFT) calculations at the CAM-B3LYP/6-311+G(d,p)//B3LYP/6-31G(d) level of theory suggest that the bowl-to-bowl inversion passes through a planar transition state with the activation barrier of 15.1 kcal mol⁻¹ (Figure S24). This value is smaller than the calculated rotation barrier of the peripheral mesityl group (26.1 kcal mol⁻¹) (Figure S25).

Figure S24. Calculated inversion behavior of **3** (CAM-B3LYP/6-311+G(d,p)//B3LYP/6-31G(d)).

Figure S25. Calculated rotation behavior of the mesityl group of **9** (CAM-B3LYP/6-311+G(d,p)//B3LYP/6-31G(d)).

4. The biradical structure for compound **3** would provide a structure with four pi-sextets compared with the neutral compound having three pi-sextets. According to Clar's rule, the diradical would be a plausible system (see figure). However, the ^1H NMR spectra of **3** does not show any significant broadening of the signals. Nevertheless, a further comment could be given in the main manuscript.

Response: We appreciate the essential comment from this reviewer. The non-dominant contribution of an open-shell diradical state would be due to a small energetical gain given by the formation of one Clar's sextet, as seen in the closed shell nature of *p*-quinodimethane and the related molecules. This description has been added to the main text (page 11, line 15) as follows.

These results suggest the non-dominant contribution of an open-shell diradical state, which will be due to a small energetical gain given by the formation of one Clar's sextet in its resonance structure (Figure S35)⁷¹.

Figure S35. Possible contributing structures in the resonance hybrid of **3**.

5. The experimental findings match pretty well with the theoretical calculations. In particular, the electrostatic potential shows the charge distribution along the molecule. Furthermore, in the ACID study the paratropic current of five member rings is also observed. However, a comment on this fact should be added to the discussion.

Response: We thank this reviewer for his/her fruitful advice. We rationalized the paratropic ring current at the five membered rings as local antiaromaticity due to cyclopentadienyl cation-like electronic contribution. The detailed explanation has been included in the main text (page 11, line 2) as follows.

Besides, the ACID calculations predict counter-clockwise ring current at the five-membered rings, which accord with the positively large NICS values. Although the exact reason remains unclear, this observation can be accounted for by the local antiaromaticity at the five membered rings, which could be due to the cyclopentadienyl cation-like electronic contribution associated with the electron-deficient character of *as*-indacenopyrene segment.

6. In addition to the isotropic NICS(0), calculation of NICS(1)_{zz} (out-of-plane tensor component computed 1 Å above), since typically performs better than isotropic NICS(0). See: H. Fallah-Bagher-Shaidaei, C. S. Wannere, C. Corminboeuf, R. Puchta and P. v. R. Schleyer, *Org. Lett.*, 2006, 8, 863).

Response: We thank this reviewer for his/her fruitful suggestion. The NICS(1)_{zz} calculations were conducted (Figure 4c and Table S2). The results indicates an essentially identical trend observed for NICS(0)_{iso} values, which again supports our assumption that the contribution of macrocyclic antiaromaticity is negligible in the electronic structure of *as*-indacenoterrylene **3**. This description has been added to the main text (page 10, line 26) as follows.

An essentially similar trend was also observed for the NICS(1)_{zz} values (Figure 4c and Table S2)⁶⁹.

Figure 4c. NICS values of **3**. d, ACID plot of **3**.

Table S2. NICS values of **3**.

	NICS _{iso}			NICS _{zz}		
	NICS(0)	NICS(1) _{convex}	NICS(1) _{concave}	NICS(0)	NICS(1) _{convex}	NICS(1) _{concave}
a	-3.94	-9.39	-8.80	6.85	-10.81	-7.98
b	-2.78	-6.61	-7.23	9.59	-6.49	-7.54
c	19.54	8.13	10.24	77.12	35.46	52.32
d	6.40	2.23	0.30	35.93	14.43	18.15
e	-0.66	-4.53	-6.82	13.54	0.83	2.22
f	-2.19	-1.89	-7.26	16.68	-1.13	-6.10
g	0.99	-1.89	-3.85	22.22	3.33	2.15

- In figure 2 parts (a, b, c) are not clearly separated.

Response: We appreciate the comment from this reviewer. We modified Figure 2 according to this suggestion.

Figure 2.

- Some NMR spectra in the supporting information are not intense enough to determine the purity of the compounds. Despite the spectra have been amplified, but only the aromatic region.

Response: We thank this reviewer for his/her careful checking. We have enlarged several spectra (Figure S7, S12, S13, S15, and S17).

- In the first introducing paragraph some references are provided but some recent ones, as claimed, should be added. See, for instance: J. Am. Chem. Soc. 2018, 140, 17188-17196; Angew. Chem. Int. Ed. 2017, 56, 9003-9007; for a recent review: Commun Chem 2019, 2, 58). The same for supramolecular interactions, see, for instance: Nat. Commun. 2015, 6:8215; Angew. Chem. Int. Ed. 2014, 53, 2170-2175.

Response: We sincerely thank this reviewer for his/her important suggestions. Our manuscript certainly lacked some important pieces in this research field. We have cited these papers in the revised manuscript (ref. 8, 13, 14, 16, and 17).

Reviewer #3

The submitted work by Tanaka, Fukui and Shinokubo presents the synthesis and characterization of as-indacenoterrylene. This molecular system is stable and exhibits a low HOMO-LUMO gap due to intramolecular charge transfer (CT). The work provides novel avenues for the design of stable hydrocarbons with appealing optoelectronic properties. In general, the manuscript is well written, clear and properly organized. In my opinion, this is a nice piece of work that deserves the attention of the Nature Communications' readers. Despite that, several issues need addressed before I can recommend it for publication.

Response: We express our sincere thanks to this reviewer for his/her extensive reading and we are so delighted to see these supportive comments.

1. The authors claim that the near-infrared absorption band corresponds to an intramolecular charge transfer (CT) excitation, but in my opinion the nature of this transition needs to be further justified. Looking at the main orbital contribution (HOMO-to-LUMO in Table S1) and the large delocalization of the orbitals (Figures 4 and S27), it does not seem to me that there is a strong CT character. Moreover, electrostatic potential map (Figure 4) does not provide an unambiguous proof of the nature of the transition. This is a central element of the study and one of the main conclusions, thus, additional evidence for the (intra) CT character should be provided. For instance, the solvent polarity dependence of the absorption band might be an indication of CT. In parallel, the authors should present computational evidence of the CT nature. Simple comparison between Mulliken, NPA and/or other schemes should confirm different ground and excited state charge localization. Representation of natural transition orbitals, or more sophisticate analysis based on attachment/detachment densities [J. Am. Chem. Soc. 131 (2009) 3913-3922], the transition density matrix [J. Chem. Theory Comput. 8 (2012) 2777-2789] or the hole/electron overlap [J. Chem. Phys. 128 (2008) 044118] and/or others could be also employed.

Response: We would like to express our gratitude for his/her valuable suggestions. We agree with the opinion that further evidences to support the CT character are essential.

(i) We examined the solvent effect on the absorption spectra of mesityl-substituted derivative **9** (Figure S27). The spectra varied slightly by the solvent but did not show clear dependence on the solvent polarity. We attribute these results to the less-organized solvent arrangement surrounding the electrically neutral ground state. Indeed, even apparent donor-accepter dyes show small solvent effect in their absorption spectrum as different from their emission spectrum (ref. 63, 64). On the other hand, the ill-defined absorption of **3** and **9** with the absence of vibrational bands in the NIR region is characteristic of CT transition, which should be an experimental evidence to support our hypothesis. This description has been added to the main text (page 10, line 14, 19) as follows.

Although the solvent effect on the absorption spectra of mesityl-substituted derivative **9** is negligible (Figure S27), this observation can be rationalized in terms of less-organized solvent arrangement surrounding the electrically neutral ground state^{65,66}. Even apparent donor-accepter dyes show small solvent effect in their absorption as different from their emission^{63,64}.

Figure S27. UV/vis/NIR absorption spectra of **3** and **9** in CH_2Cl_2 .

(ii) We have also calculated the S_1 state of **3** and checked the electrostatic potential map (Figure 4b). The calculations predict that the S_1 state has been rather electrostatically neutralized, which supports the CT character in the S_0 - S_1 transition. This description has been added to the main text (page 10, line 18) as follows.

Figure 4b. Electrostatic potential map of S_0 - and S_1 -states of **3**.

(iii) Furthermore, we conducted the natural transition orbital analysis for **3** (Figure S33). The results indicate that the S_0 - S_1 transition accompanies with the change of orbital

distribution from the terrylene segment to the *as*-indacenopyrene segment, which again supports our hypothesis. This description has been added to the main text (page 10, line 13) as follows.

This assumption has also been supported by the natural transition orbital analysis (Figure S33). Indeed, the ill-defined absorption feature of **3** and **9** with the absence of vibrational bands in the NIR region is characteristic of CT transition^{63,64}.

Figure S33. Natural transition orbital analysis of *as*-indacenoterrylene **3** (CAM-B3LYP/6-311+G(d,p), isovalue = 0.03).

2. On the other hand, the authors suggest that, despite the small HOMO-LUMO gap, *as*-indacenoterrylene holds no diradical character. This conclusion is based (in part) on CASSCF(2,2)/6-31G calculations. This level of computation is rather low, especially considering the size of the molecule, which allows the use of higher order approximations. Is it possible to increase the active space of the calculation and the basis set at least to include polarization functions (6-31G*)?

Response: We thank this reviewer for his/her constructive advice. According to the reviewer's suggestion, we have conducted the CASSCF calculations at the CASSCF(4,6)/6-31G(d) level. The calculation also gave a relatively small diradical index of $y = 0.07$. These results have been included in the main text (page 11, line 9) as follows.

The increase of the calculation level (CASSCF(4,6)/6-31G(d)) again provided a relatively small diradical index of $y = 0.07$.

Moreover, as far as I understand, these calculations have been performed on a closed-shell (CS) optimized geometry, as obtained with restricted Kohn-Sham (RKS) calculations, which sometimes might impose a CS solution (even at CASSCF level). Hence, in order to solidify the paper conclusions, I would suggest to try to optimize the open-shell (OS) solution and the triplet state. What is the OS/CS gap? And the relative energy to the lowest triplet? What is the diradical index (y) at the OS geometry (if found)? Stability analysis of the CS solution with respect to the OS singlet (e.g., RKS vs. UKS), might also help.

Response: According to the reviewer's suggestion, we have re-optimized the geometries of **3** with closed-shell (CS) singlet, open-shell (OS) singlet and triplet states at the (U)CAM-B3LYP/6-311+G(d,p) level. The OS singlet geometry is structurally almost close to that of the CS singlet structure and more stable than the CS singlet by 0.9 kcal mol⁻¹. The stability of the OS singlet solution has been checked with the stability analysis. The diradical index at the optimized OS singlet has been calculated to be $y = 0.03$. The triplet state is energetically unfavourable with the energy gap ΔE_{ST} of 6.7 kcal mol⁻¹. These results suggest the non-dominant contribution of an open-shell diradical state. This description has been included to the main text (page 11, line 10) as follows.

In addition, the closed-shell (CS) singlet, open-shell (OS) singlet and triplet states of **3** were optimized at the (U)CAM-B3LYP/6-311+G(d,p) level (Table S3–5). The OS singlet state is structurally almost identical with the CS singlet state, being slightly more stable than the CS singlet state by 0.9 kcal mol⁻¹ with the small diradical index of $y = 0.03$. The triplet state is energetically unfavourable with the energy gap ΔE_{ST} of 6.7 kcal mol⁻¹.

Page 9. Remove DFT from the sentence: "DFT calculations of **3** at the CASSCF(2,2)/6-31G level".

Response: We thank this reviewer for his/her careful checking. We have omitted the word "DFT" in this sentence.

Page 10. Figure 4. The authors show the similarities between the HOMO of **3** and terrylene, and the match between the LUMO of **3** and *as*-indacenopyrene. This is actually a very nice and simple comparison. But, I'm not sure it can be associated to a CT transition. What about the LUMO of terrylene? Does it resemble the LUMO of **3** as well? And the HOMO of *as*-indacenopyrene?

Response: We thank this reviewer for his/her fruitful advice. We have included the comparison of HOMO and LUMO of *as*-indacenoterrylene **3**, *as*-indacenopyrene, and terrylene (Figure S32). The LUMO of **3** is energetically much lower than LUMO of terrylene by 0.7 eV. The distribution and energy level of HOMO of **3** are almost identical with those of the HOMO of terrylene rather than those of HOMO of *as*-indacenopyrene. This description has been added to the main text (page 10, line 5) as follows.

The calculated HOMOs and LUMOs of **3** and its fragments, *as*-indacenopyrene and terrylene, are shown in Figure 4a and Figure S32. The HOMO of **3** is delocalized along the

longitudinal axis, whose distribution and energy level are almost identical to those of the HOMO of terrylene rather than those of HOMO of *as*-indacenopyrene. In contrast, the LUMO of **3** is mainly located at the centre of the molecule, which resembles the LUMO of *as*-indacenopyrene and energetically much lower than the LUMO of terrylene by 0.7 eV.

Figure 4a. HOMO and LUMO of **3**. Inset: HOMO of terrylene and LUMO of *as*-indacenopyrene.

Figure S32. Comparison of frontier orbitals of *as*-indacenoterrylene **3**, *as*-indacenopyrene and terrylene.

REVIEWER COMMENTS

Reviewer #1 (Remarks to the Author):

The authors have properly addressed all issues raised by me. It is now ready to publish in NC. Congratulations!

Reviewer #2 (Remarks to the Author):

The authors have successfully addressed all the concerns from my side. Actually, they have given convincing arguments and from my viewpoint, they have also responded to the other reviewers who have raised smart questions, again with a nice answers.

Thus, I feel that the manuscript in its present form meets the criteria to be accepted for publication in Nat. Commun. basically as it stands.

Reviewer #3 (Remarks to the Author):

The authors have properly tackled several of the issues I raised in my first review of the manuscript, but there are still some points that need to be addressed. In particular, the authors claim that S1 state holds a strong intramolecular CT character. Based on the experimental data and computational results provided, I don't think that this statement is justified. The authors need to properly characterize the nature of the lowest singlet transition and find an alternative rationalization for the measured and computed low transition energy.

Since this is a central element of the study, at this stage, I cannot recommend the present work for publication in Nature Communications.

See attached document (review_ncomms_20_15443A.pdf) for more details.

Sincerely,
David Casanova

Review of NCOMMS-20-15443A

Reviewer 3 (comments to authors' response in blue)

The submitted work by Tanaka, Fukui and Shinokubo presents the synthesis and characterization of as-indacenoterrylene. This molecular system is stable and exhibits a low HOMO-LUMO gap due to intramolecular charge transfer (CT). The work provides novel avenues for the design of stable hydrocarbons with appealing optoelectronic properties.

In general, the manuscript is well written, clear and properly organized. In my opinion, this is a nice piece of work that deserves the attention of the Nature Communications' readers. Despite that, several issues need addressed before I can recommend it for publication.

Response: We express our sincere thanks to this reviewer for his/her extensive reading and we are so delighted to see these supportive comments.

1. The authors claim that the near-infrared absorption band corresponds to an intramolecular charge transfer (CT) excitation, but in my opinion the nature of this transition needs to be further justified. Looking at the main orbital contribution (HOMO-to-LUMO in Table S1) and the large delocalization of the orbitals (Figures 4 and S27), it does not seem to me that there is a strong CT character. Moreover, electrostatic potential map (Figure 4) does not provide an unambiguous proof of the nature of the transition. This is a central element of the study and one of the main conclusions, thus, additional evidence for the (intra) CT character should be provided. For instance, the solvent polarity dependence of the absorption band might be an indication of CT. In parallel, the authors should present computational evidence of the CT nature. Simple comparison between Mulliken, NPA and/or other schemes should confirm different ground and excited state charge localization. Representation of natural transition orbitals, or more sophisticate analysis based on attachment/detachment densities [J. Am. Chem. Soc. 131 (2009) 3913-3922], the transition density matrix [J. Chem. Theory Comput. 8 (2012) 2777-2789] or the hole/electron overlap [J. Chem. Phys. 128 (2008) 044118] and/or others could be also employed.

Response: We would like to express our gratitude for his/her valuable suggestions. We agree with the opinion that further evidences to support the CT character are essential.

(i) We examined the solvent effect on the absorption spectra of mesityl-substituted derivative **9** (Figure S27). The spectra varied slightly by the solvent but did not show clear dependence on the solvent polarity. We attribute these results to the less-organized solvent arrangement surrounding the electrically neutral ground state. Indeed, even apparent donor-accepter dyes show small solvent effect in their absorption spectrum as different from their emission spectrum (ref. 63, 64). On the other hand, the ill-defined absorption of **3** and **9** with the absence of vibrational bands in the NIR region is characteristic of CT transition, which should be an experimental evidence to support our hypothesis. This description has been added to the main text (page 10, line 14, 19) as follows.

Although the solvent effect on the absorption spectra of mesityl-substituted derivative **9** is negligible (Figure S27), this observation can be rationalized in terms of less-organized solvent arrangement surrounding the electrically neutral ground state^{65,66}. Even apparent donor-accepter dyes show small solvent effect in their absorption as different from their emission^{63,64}.

Figure S27

Comment. It is true that absorption energies might not be strongly influenced by the solvent polarity, even if the state holds some CT character, while the nature of the state is more clearly manifested by the shift of emission energies. On the other hand, it is surprising that absorption profiles in different solvents for the so-called CT absorption band are identical (Figure S27).

(ii) We have also calculated the S1 state of **3** and checked the electrostatic potential map (Figure 4b). The calculations predict that the S1 state has been rather electrostatically neutralized, which supports the CT character in the S0-S1 transition. This description has been added to the main text (page 10, line 18) as follows.

Figure 4b

(iii) Furthermore, we conducted the natural transition orbital analysis for **3** (Figure S33). The results indicate that the S0-S1 transition accompanies with the change of orbital distribution from the terylene segment to the *as*-indacenopyrene segment, which again supports our hypothesis. This description has been added to the main text (page 10, line 13) as follows.

This assumption has also been supported by the natural transition orbital analysis (Figure S33).

Comment. I don't agree that the electrostatic potential map supports CT character of the S0-S1 transition. Clearly, the hole/electron natural transition orbitals exhibit large spatial overlap. Hence, from these results it is not appropriate to assign S1 as an intramolecular CT state.

Indeed, the ill-defined absorption feature of **3** and **9** with the absence of vibrational bands in the NIR region is characteristic of CT transition^{63,64}.

Figure S33

Comment. It is true that the absence of vibrational bands in the NIR region is characteristic of CT states, but the presence of this band does not univocally imply CT character. Low energy transitions in diradicals also exhibit such behavior.

2. On the other hand, the authors suggest that, despite the small HOMO-LUMO gap, as-indacenoterrylene holds no diradical character. This conclusion is based (in part) on CASSCF(2,2)/6-31G calculations. This level of computation is rather low, especially considering the size of the molecule, which allows the use of higher order approximations. Is it possible to increase the active space of the calculation and the basis set at least to include polarization functions (6-31G*)?

Response: We thank this reviewer for his/her constructive advice. According to the reviewer's suggestion, we have conducted the CASSCF calculations at the CASSCF(4,6)/6-31G(d) level. The calculation also gave a relatively small diradical index of $y = 0.07$. These results have been included in the main text (page 11, line 9) as follows.

The increase of the calculation level (CASSCF(4,6)/6-31G(d)) again provided a relatively small diradical index of $y = 0.07$.

Comment. In my opinion, the authors have properly addressed this issue. The new results consistently indicate no (or very weak) diradical character of the ground state of **3**.

Moreover, as far as I understand, these calculations have been performed on a closed-shell (CS) optimized geometry, as obtained with restricted Kohn-Sham (RKS) calculations, which sometimes might impose a CS solution (even at CASSCF level). Hence, in order to solidify the paper conclusions, I would suggest to try to optimize the open-shell (OS) solution and the triplet state. What is the OS/CS gap? And the relative energy to the lowest triplet? What is the diradical index (y) at the OS geometry (if found)? Stability analysis of the CS solution with respect to the OS singlet (e.g., RKS vs. UKS), might also help.

Response: According to the reviewer's suggestion, we have re-optimized the geometries of **3** with closed-shell (CS) singlet, open-shell (OS) singlet and triplet states at the (U)CAM-B3LYP/6-311+G(d,p) level. The OS singlet geometry is structurally almost close to that of the CS singlet structure and more stable than the CS singlet by $0.9 \text{ kcal mol}^{-1}$. The stability of the OS singlet solution has been checked with the stability analysis. The diradical index at the optimized OS singlet has been calculated to be $y = 0.03$. The triplet state is energetically unfavourable with the energy gap ΔE_{ST} of $6.7 \text{ kcal mol}^{-1}$. These results suggest the non-dominant contribution of an open-shell diradical state. This description has been included to the main text (page 11, line 10) as follows.

In addition, the closed-shell (CS) singlet, open-shell (OS) singlet and triplet states of **3** were optimized at the (U)CAM-B3LYP/6-311+G(d,p) level (Table S3–5). The OS singlet state is structurally almost identical with the CS singlet state, being slightly more stable than the CS singlet state by $0.9 \text{ kcal mol}^{-1}$ with the small diradical index of $y = 0.03$. The triplet state is energetically unfavourable with the energy gap ΔE_{ST} of $6.7 \text{ kcal mol}^{-1}$.

Comment. In my opinion, the authors have properly addressed this issue. The new results consistently indicate no (or very weak) diradical character of the ground state of **3**.

Page 9. Remove DFT from the sentence: "DFT calculations of **3** at the CASSCF(2,2)/6-31G level".

Response: We thank this reviewer for his/her careful checking. We have omitted the word "DFT" in this sentence.

Comment. In my opinion, the authors have properly addressed this issue.

Page 10. Figure 4. The authors show the similarities between the HOMO of **3** and terrylene, and the match between the LUMO of **3** and *as*-indacenopyrene. This is actually a very nice and simple comparison. But, I'm not sure it can be associated to a CT transition. What about the LUMO of terrylene? Does it resemble the LUMO of **3** as well? And the HOMO of *as*-indacenopyrene?

Response: We thank this reviewer for his/her fruitful advice. We have included the comparison of HOMO and LUMO of *as*-indaceno-terrylene **3**, *as*-indacenopyrene, and terrylene (Figure S32). The LUMO of **3** is energetically much lower than LUMO of terrylene by 0.7 eV. The distribution and energy level of HOMO of **3** are almost identical with those of the HOMO of terrylene rather than those of HOMO of *as*-indacenopyrene. This description has been added to the main text (page 10, line 5) as follows.

The calculated HOMOs and LUMOs of **3** and its fragments, *as*-indacenopyrene and terrylene, are shown in Figure 4a and Figure S32. The HOMO of **3** is delocalized along the longitudinal axis, whose distribution and energy level are almost identical to those of the HOMO of terrylene rather than those of HOMO of *as*-indacenopyrene. In contrast, the LUMO of **3** is mainly located at the centre of the molecule, which resembles the LUMO of *as*-indacenopyrene and energetically much lower than the LUMO of terrylene by 0.7 eV.

Figures 4a and S32

Comment. The new results show that the HOMO and LUMO of **3** are obtained as orbital mixings of *as*-indacenopyrene and terrylene, resulting in a HOMO with an energy close to the terrylene HOMO, while the LUMO of **3** lies between *as*-indacenopyrene and terrylene. In other words, there is a sizeable stabilization of the LUMO with respect to terrylene (while the HOMO energy barely changes). As a result, the lowest transition in **3** is expected to be lower than in terrylene. Maybe the comparison of S1 energies (and transition dipole moments) in *as*-indacenopyrene, terrylene and **3** might help to further clarify the nature of the lowest excited singlet.

General comment. Although the use of *as*-indacenopyrene and terrylene frontier orbitals and energies to describe and characterize the S₀ to S₁ transition in **3** is a nice approach that brings a “simple” rationalization of the nature of S₁, it cannot be used here as to label the lowest singlet as a CT state. The description of the transition in terms of two fragments that largely overlap, i.e. HOMO and LUMO with similar spatial distributions, cannot be used to assign CT character to the transition. In other words, I barely see any charge separation in it. Therefore, in my opinion, the authors should avoid to the term “intramolecular CT” and should find an alternative explanation for the NIR absorption energy measured and computed. This is a central aspect of the present study that needs to be addressed before the manuscript can be accepted for publication.

Response to Reviewers

Reviewer #1

The authors have properly addressed all issues raised by me. It is now ready to publish in NC.

Congratulations!

Response: We express our gratitude to this reviewer for his/her extensive reading and we are delighted to see the recommendation for acceptance.

Reviewer #2

The authors have successfully addressed all the concerns from my side. Actually, they have given convincing arguments and from my viewpoint, they have also responded to the other reviewers who have raised smart questions, again with a nice answers.

Thus, I feel that the manuscript in its present form meets the criteria to be accepted for publication in *Nat. Commun.* basically as it stands.

Response: We express our gratitude to this reviewer for his/her extensive reading and we are delighted to see the recommendation for acceptance.

Reviewer #3

The authors have properly tackled several of the issues I raised in my first review of the manuscript, but there are still some points that need to be addressed. In particular, the authors claim that S1 state holds a strong intramolecular CT character. Based on the experimental data and computational results provided, I don't think that this statement is justified. The authors need to properly characterize the nature of the lowest singlet transition and find an alternative rationalization for the measured and computed low transition energy.

Since this is a central element of the study, at this stage, I cannot recommend the present work for publication in Nature Communications.

General comment. Although the use of *as*-indacenopyrene and terrylene frontier orbitals and energies to describe and characterize the S₀ to S₁ transition in **3** is a nice approach that brings a “simple” rationalization of the nature of S₁, it cannot be used here as to label the lowest singlet as a CT state. The description of the transition in terms of two fragments that largely overlap, i.e. HOMO and LUMO with similar spatial distributions, cannot be used to

assign CT character to the transition. In other words, I barely see any charge separation in it. Therefore, in my opinion, the authors should avoid to the term “intramolecular CT” and should find an alternative explanation for the NIR absorption energy measured and computed. This is a central aspect of the present study that needs to be addressed before the manuscript can be accepted for publication.

Response: We express our gratitude to this reviewer for his/her extensive reading. We appreciate the critical comments from this reviewer.

We understand the concern from this reviewer on our discussion of the intramolecular CT character. We also agree with the opinion that CT character is not the principal origin to explain the electronic structure of *as*-indacenoterrylene. In our opinion, the notable feature of the current study is that the electronic nature of *as*-indacenoterrylene cannot be explained in terms of conventional concepts: antiaromaticity and biradical character. However, we have to admit that our original manuscript included several overstatements on the intramolecular CT character.

We sought an alternative explanation for the the electronic structure of *as*-indacenoterrylene. One plausible description could be the coexistence of *as*-indacenopyrene- and terrylene-units, which are moderately merged together without diminishing the individual electronic character. Because *as*-indacenopyrene and terrylene segments are electron-accepting and electron-donating, respectively, the S_0 - S_1 transition of *as*-indacenoterrylene would include some contribution of the CT character but is not the direct origin of the NIR absorption. Accordingly, we have revised the manuscript to downplay the descriptions on the CT character.

The changes are as follows.

1. In page 1, line 16, we have deleted the following sentence emphasizing the CT character in the S_0 - S_1 transition.

The low-energy absorption was attributed to the intramolecular charge-transfer from the electron-rich terrylene segment to the electron-deficient *as*-indacenopyrene segment.

2. In page 2, line 23, we have deleted the sentence relating to the charge-transfer character. Instead, we have added new sentences to emphasize that the electronic nature of *as*-indacenoterrylene cannot be explained in terms of antiaromaticity and biradical character as follows.

Before: The low-energy absorption was attributed to the intramolecular charge-transfer from the electron-rich terrylene segment to the electron-deficient *as*-indacenopyrene segment.

After: *as*-Indacenoterrylene **3** is stable enough even without any peripheral substituents. This feature is contrasting with other NIR-absorbing hydrocarbons such as singlet diradicaloids and antiaromatic molecules.

3. We have eliminated the term of "Charge-transfer character" from Figure 1.

4. In page 8, line 22, the comparison with previously reported all-carbon donor-acceptor systems has been removed.

5. In page 10, line 11, we have omitted the following sentences explaining the electronic structure of *as*-indacenoterrylene in term of charge-transfer character.

These results suggest that the low-energy absorption of *as*-indacenoterrylene **3** is due to the intramolecular charge-transfer character of its HOMO–LUMO transition. This assumption has also been supported by the natural transition orbital analysis (Figure S33). Indeed, the ill-defined absorption feature of **3** and **9** with the absence of vibrational bands in the NIR region is characteristic of CT transition^{63,64}. This notion is corroborated by the electrostatic potential map of **3**, in which the central *as*-indacenopyrene segment is positively charged relative to the surrounding fused naphthalene units (Figure 4b). Importantly, this charge deviation becomes neutralized in the S₁ state. The Although the solvent effect on the absorption spectra of mesityl-substituted derivative **9** is negligible (Figure S27), this observation can be rationalized in terms of less-organized solvent arrangement surrounding the electrically neutral ground state^{65,66}. Even apparent donor-accepter dyes show small solvent effect in their absorption as different from their emission^{63,64}.

6. In page 11, line 22, we have included a new paragraph to explain the electronic structure of *as*-indacenoterrylene as follows.

The above-mentioned studies indicate that the electronic structure of *as*-indacenoterrylene **3** cannot be satisfactorily explained in terms of its antiaromaticity and diradical character. One plausible description could be the coexistence of *as*-indacenopyrene- and terrylene-units, which are moderately merged together without diminishing the individual electronic character. TD-DFT calculations predict that, while the S₀–S₁ transition of terrylene is allowed (oscillator strength = 0.8363) with a large gap of 2.52 eV, that of *as*-indacenopyrene is forbidden (oscillator strength = 0.0118) with a small gap of 0.97 eV

(Figure S33, S34). The oscillator strength and energy gap of **3** (0.1226 and 1.51 eV) lie between those of terrylene and *as*-indacenopyrene. Importantly, the transition dipoles of **3**, *as*-indacenopyrene and terrylene are directed in the same molecular axis, which could be the origin of the moderate mixing of the transitions. Because of the coexistence of electron-accepting and electron-donating segments⁶³, the S₀–S₁ transition of **3** could include some contribution of the charge-transfer (CT) character. Indeed, the electrostatic potential maps suggest that the S₀–S₁ transition associates with the change of orbital- and charge-distributions (Figure S30, S32). The ill-defined absorption feature of **3** and **9** with the absence of vibrational bands in the NIR region is reminiscent of CT character.

7. The electrostatic potential maps of S₀- and S₁-states have been removed from Figure 4. Accordingly, the caption has been modified.

8. In page 12, line 10, we have eliminated the discussion relating to the charge-transfer character and modified the discussion part as follows.

Before: This low-energy absorption was attributed to the intramolecular charge-transfer character of the HOMO–LUMO transition, which is due to the coexistence of an electron-rich terrylene segment and an electron-deficient *as*-indacenopyrene segment. The absorption tail of **3** is not only comparable to organic diradicaloid **1** and antiaromatic hydrocarbon **2**, but also more red-shifted than those of other donor–acceptor systems that consist exclusively of carbon and hydrogen atoms. Importantly, *as*-indacenoterrylene **3** exhibits outstanding stability despite the absence of peripheral substituents, which stands in sharp contrast to the behaviour of other NIR-absorbing hydrocarbons such as biradicaloids and antiaromatic molecules. The current study thus offers new design guidelines for curved aromatic hydrocarbons with narrow HOMO–LUMO gaps and all-carbon donor–acceptor systems.

After: The absorption tail of **3** is comparable to organic diradicaloid **1** and antiaromatic hydrocarbon **2**. However, *as*-indacenoterrylene **3** exhibits outstanding stability despite the absence of peripheral substituents, which stands in sharp contrast to the behaviour of other NIR-absorbing hydrocarbons such as biradicaloids and antiaromatic molecules. The theoretical studies suggest that the electronic nature of **3** cannot be satisfactorily explained in terms of antiaromaticity and biradical character. The current study thus offers new design guidelines for aromatic hydrocarbons with narrow HOMO–LUMO gaps.

Comment. It is true that absorption energies might not be strongly influenced by the solvent polarity, even if the state holds some CT character, while the nature of the state is more clearly manifested by the shift of emission energies. On the other hand, it is surprising that

absorption profiles in different solvents for the so-called CT absorption band are identical (Figure S27).

Response: We appreciate the critical comment from this reviewer. We conducted the further literature search and found two CT dyes exhibiting almost identical absorption spectra in spite of the change of the solvent polarity (S. Yamaguchi *et al.*, *J. Am. Chem. Soc.* **2006**, *128*, 15934; H.-B. Sun *et al.*, *J. Phys. Chem. B* **2010**, *114*, 128.). For example, a (donor-acceptor-donor)-type oligo(phenyleneethynylene) reported by Yamaguchi and co-workers displayed absorption maxima at 464 nm (cyclohexane), 465 nm (benzene), 454 nm (THF), and 454 nm (MeOH). We consider that the symmetric structure would be one reason for the negligible solvent-dependence. However, we have already downplayed the descriptions relating to the CT character in the main text and deleted the discussion about the solvent dependence.

Comment. I don't agree that the electrostatic potential map supports CT character of the S_0 - S_1 transition. Clearly, the hole/electron natural transition orbitals exhibit large spatial overlap. Hence, from these results it is not appropriate to assign S_1 as an intramolecular CT state.

Response: We agree with the opinion that the electrostatic potential maps do not strongly support the charge-transfer character. The potential maps have been removed from Figure 4. Instead, they have been included in SI as supporting materials to evaluate the electronic structure of *as*-indacenoterrylene.

Comment. It is true that the absence of vibrational bands in the NIR region is characteristic of CT states, but the presence of this band does not univocally imply CT character. Low energy transitions in diradicals also exhibit such behavior.

Response: We thanks this reviewer for his/her important opinion. We also agree that the broad absorption should not be a crucial evidence to suggest strong charge-transfer character. However, the diradical character of *as*-indacenoterrylene **3** is rather marginal. There would be an alternative explanation for the the absence of vibrational bands.

Comment. The new results show that the HOMO and LUMO of **3** are obtained as orbital mixings of *as*-indacenopyrene and terrylene, resulting in a HOMO with an energy close to the terrylene HOMO, while the LUMO of **3** lies between *as*-indacenopyrene and terrylene. In other words, there is a sizeable stabilization of the LUMO with respect to terrylene (while the HOMO energy barely changes). As a result, the lowest transition in **3** is expected to be lower than in terrylene. Maybe the comparison of S_1 energies (and transition dipole

moments) in *as*-indacenopyrene, terrylene and **3** might help to further clarify the nature of the lowest excited singlet.

Response: We thank this reviewer for his/her helpful advise. According to the suggestion, TD-DFT calculations were conducted for *as*-indacenopyrene and terrylene. While the S_0-S_1 transition of terrylene is allowed (oscillator strength = 0.8363) with a large gap of 2.52 eV, that of *as*-indacenopyrene is forbidden (oscillator strength = 0.0118) with a small gap of 0.97 eV (Figure S34, S35). The oscillator strength and energy gap of **3** (0.1226 and 1.51 eV) lie between those of terrylene and *as*-indacenopyrene. Importantly, the transitions dipoles of **3**, *as*-indacenopyrene and terrylene are directed in the same molecular axis. These results encouraged us to propose the moderate mixing of electronic natures of *as*-indacenopyrene and terrylene as an alternative explanation as follows (page 11, line 22).

The above-mentioned studies indicate that the electronic structure of *as*-indacenoterrylene **3** cannot be satisfactorily explained in terms of its antiaromaticity and diradical character. One plausible description could be the coexistence of *as*-indacenopyrene- and terrylene-units, which are moderately merged together without diminishing the individual electronic character. TD-DFT calculations predict that, while the S_0-S_1 transition of terrylene is allowed (oscillator strength = 0.8363) with a large gap of 2.52 eV, that of *as*-indacenopyrene is forbidden (oscillator strength = 0.0118) with a small gap of 0.97 eV (Figure S33, S34). The oscillator strength and energy gap of **3** (0.1226 and 1.51 eV) lie between those of terrylene and *as*-indacenopyrene. Importantly, the transition dipoles of **3**, *as*-indacenopyrene and terrylene are directed in the same molecular axis, which could be the origin of the moderate mixing of the transitions.

REVIEWERS' COMMENTS:

Reviewer #3 (Remarks to the Author):

The authors have properly addressed all the issues I raised in my previous review. In my opinion the manuscript should be accepted for publication in Nature Communications in its present form.